# To Simulate Neural Organoid: A Framework and A Benchmark based on AI

## Abstract

Neural organoids hold significant value for AI modeling, cognitive exploration, and medical discovery in academic research. However, the current research on neural organoids primarily relies on trial-and-error experimental design methods that are time-consuming and expensive. Moreover, due to the intrinsic unknowability of complex biological systems, purely rational deduction through mathematical and physical modeling is nearly impossible. As a result, the design of organoid experiments is constrained by the above limitations. With AI models being applied to address diverse biological challenges, the demand for novel experimental paths for neural organoids has become urgent. In response to the above issues, we propose the first neural organoid simulation framework to realistically reconstruct various details of interaction experiments using real-world mature organoids. This framework employs advanced neural computing models as elements, harnessing AI methods to enable stimulation, response, and learning functionalities. The significant consumption can be mitigated through the combination of the framework with real-world experiments. An intelligent expansion platform is also established based on spiking neural network to facilitate the exploration of organoid-machine collaborative intelligence. In addition, we introduce a benchmark for evaluating our framework, including a set of real-world organoid experimental data and a series of evaluation metrics. The experimental results show that our simulation framework features outstanding simulation capabilities and reflects similarity with real-world organoid experiments in many aspects. With the intelligent expansion platform, the performance of the combination is comparable to pure AI algorithms in a basic classification task.

## 1 Introduction

Organoids refer to in vitro cultured cell tissues with structure and functions. They are generated through the application of stem cell-directed differentiation techonology. So far, various organs have been successfully cultured in vitro, including the heart (Voges et al., 2017), liver (Huch et al., 2013), lung (Wilkinson et al., 2017), gut (Spence et al., 2011), brain (Paşca et al., 2015; Qian et al., 2016), and etc. Among them, neural organoids are essential for disease intervention (Marotta et al., 2020), cognitive and intelligence formation (Goddard et al., 2023). Some research focuses on the intelligence performance of organoids themselves (Smirnova et al., 2023; Kagan et al., 2022) and start to investigate the organoid-machine collaborative intelligence. Furthermore, the third generation of neural network, Spiking Neural Network (SNN) (Maass, 1997), is regarded as a promising method for interacting with biological tissues by their highly biomimetic design and low power consumption.

Due to the intricate nature of biological mechanisms, neural organoids experiments often rely on heuristics, lacking a rational design approach. Pure rational design based on mathematical models for neural organoids is unfeasible. Consequently, researchers attempt to conduct numerous trial-and-error experiments to derive some patterns. However, conducting pure heuristic experiments imposes a huge finance and consumable load with low success rates. Pure rational design and pure experimentation are opposing extremes that both fall short of ideal research methodologies. Recently, AI for science modeling exhibited substantial effectiveness across diverse domains (Senior et al., 2020; Bi et al., 2023). Leveraging AI methods to simulate the structure and learning mechanisms of organoids fulfills the pressing need within the field for novel research methodologies.

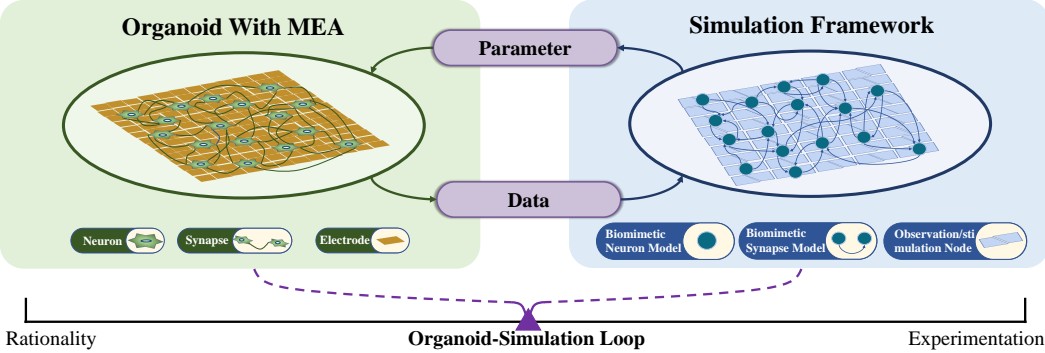

Figure 1: An illustration of Organoid-Simulation Loop.

In this work, we propose the first neural organoid simulation framework(NOSF) based on AI method, which is a novel tool for organoid research. The framework focuses on reconstructing the interaction process between mature organoid and machines without invoving culture and proliferation of neuron. We employ advanced computational neural modeling results as elements to establish an Observation/Stimulation (O/S) array, O/S information transmission mechanism and simulation organoid network that resemble real-world organoid experiments closely. To meet the requirements of exploring collaborative intelligence, our framework is equipped with an intelligent expansion platform based on SNN. We collect some data in a real-world neural organoid experiment and present a series of evaluation metrics to measure the level of simulation, which forms the first benchmark of neural organoid simulation. A comprehensive evaluation of the framework is conducted through substantial experiments, resulting in remarkable simulation results and intelligent expansion effects. In addition, the simulation framework can be used by organoid researchers to conduct pre-experiments, saving the expensive trial-and-error costs of repeating real-world experiments. Framework can be employed to perform pre-experiments, formulating the results as "Parameters" to guide real-world experiments and minimize the need for trial-and-error. After real-world experiments, the recorded data will also be fed back into the simulation framework as "Data" to improve the performance of simulation. As depicted in Fig.1, the "Parameters" and "Data" form an experimental loop, which strikes a balance between rationality and experimentation and demonstrates a novel methodology for organoid research.

Our contributions are summarized as follows: **(1)** The first neural organoid simulation framework is proposed to reconstruct most of the details in real-world organoid interaction experiments while alleviating the enormous cost of repeated real-world experiments. **(2)** An organoid intelligence expansion platform is developed using the SNN algorithm to explore organoids-machine collaborative intelligence in a novel way. **(3)** The first benchmark is proposed for organoid simulation framework, including evaluation metrics and real-world organoid experiment data. We conduct abundant experiments to evaluate the simulation similarity and the performance of intelligence expansion and ultimately achieve remarkable experimental results.

## 2 RELATED WORK

**Neural organoids culture and intelligence.** Due to ethical concerns, previous research methods generally focused on the sliced neural tissues from post-mortem corpse brain specimens. The development of induced pluripotent stem cell (iPSCs) (Thomson et al., 1998) technology enabled the culture of neural organoids in vitro. Earlier works had successful culture of multiple types of neural organoids, including the cerebellum (Muguruma et al., 2015), hippocampus (Sakaguchi et al., 2015), pituitary gland (Ozone et al., 2016), retina (Eiraku et al., 2011), and etc. Some in vitro neurons generated through iPSCs adhere closely to microelectrode array (MEA) and then proliferate until form the organoid. MEA has the capacity to monitor the electrical signal activity of neurons and can also apply stimulation to neurons regions. The utilization of neural organoids provides researchers in neuroscience and cognitive science with innovative methods to investigate the origin of intelligence. Generated from the human and rodent stem cells, Kagan et al. (2022) constructed an in vitro biological neural network, which facilitated real-time interaction and stimulus-based learning between computers and neurons within the "Pong" game. Similarly, Cai et al. (2023) propose the concept of "Brainware", which considers neural organoids as a hardware alongside silicon-based

Figure 2: From left to right, including GIF neurons, small-world connections, and AMPA synapses.

chips. They found that in vitro neural networks exhibit nonlinear dynamics and fading memory properties, representing a concrete manifestatio of intelligence.

**SNN and SNN for biological signal.** SNN algorithms are renowned for their robustness, biological interpretability, and energy efficiency in the Artificial Intelligence community. SNN adopts binary spikes as the carriers of information transmission with inherent suitability and advantages in biological neuron information processing. Some SNN models' architecture and training algorithms are inspired by traditional Artificial Neural Networks (ANN) (Kim et al., 2022; Zhu et al., 2022; Yao et al., 2022), resulting in comparable performance in computer vision, computational speech, and other domains. SEW ResNet (Fang et al., 2021) proposed a novel SNN design pattern called Spike-Element-Wise (SEW) and established a deep residual network architecture based on SNN. Transformer was incorporated into the SNN in (Yao et al., 2023) known as Spike-driven Transformer. Some studies applied SNN to electroencephalogram (EEG) and electromyogram (EMG) signals, leading the way in exploring the potential use of SNN for neural information processing. Gong et al. (2023) observed that SNN can effectively capture the complex dynamic characteristics of biological neurons. They introduced an SNN network combined with LSTM for EEG analysis and yielded outstanding performance in emotion recognition. Xu et al. (2023) developed a hybrid SNN architecture called SCNN combining a convolutional neural network for electromyographic pattern recognition, which improved 50.69% accuracy compared to the baseline.

## 3 PRELIMINARIES

### 3.1 GENERALIZED INTEGRATE-AND-FIRE NEURON

Numerous bionic computational neurons models were proposed, including the Hodgkin-Huxley(H-H) (Hodgkin & Huxley, 1952), Izhikevich (Izhikevich, 2004) and others. In our work, neurons with precise spikes and adaptation to various spiking modes (e.g., burst, bistability, etc.) are selected, rather than specific ion channels. For this reason, we opt for the Generalized Integrate-and-Fire (GIF) model as the neuron component within the framework. Specifically, we use the scheme of Mihalaş & Niebur (2009) and Teeter et al. (2018), which can be described as:

$$\frac{\mathrm{d}V(t)}{\mathrm{d}t} = \frac{1}{\tau}(RI + R\sum I_j(t) - (V(t) - V_{rest})) \tag{1}$$

where $V$ is membrane potential, $V_{rest}$ is resting potential, $R$ is membrane resistance, $I_j$ are an arbitrary number of internal currents and $I$ is external current. Once $V$ exceeds the threshold, a spike will be fired. Meanwhile, $V$ will be reset to the resting potential $V_{reset}$.

### 3.2 AMPA SYNAPSES

Chemical synapse serves as the core element in our simulation framework due to its plasticity, which is responsible for generating learning and memory functions. We select the AMPA receptor model as the foundational synaptic model for the framework, because of its characteristic of quick response. Specifically, we adopt the approach employed by Vijayan & Kopell (2012):

$$\frac{\mathrm{d}g}{\mathrm{d}t} = \alpha[Glu](1 - g) - \beta g \tag{2}$$

where $g$ represents the concentration of AMPA receptors, $\alpha$ signifies the binding rate constant between AMPA receptors and glutamic acid, while $\beta$ represents the dissociation rate constant. $[Glu]$ represents the concentration of the external neurotransmitter.

### 3.3 Small-world network

Small-world network (Watts & Strogatz, 1998) is a distinctive type of complex network that is prevalent in reality and holds substantial practical significance. Specifically, after extensive research, it has been proved that many large-scale neural networks within the brain, including those in the visual system and the brain stem, exhibit the characteristics of a small-world topology. Furthermore, at the level of computational modeling, small-world network exhibits favorable short-term memory characteristics. The small-world network is characterized by: (i) Build a regular ring lattice with $N$ nodes, each connected to $K$ neighbors. (ii) Reconnect each edge of each node in sequential order with a probability of $p$ and randomly select the target nodes for reconnection.

## 4 Method

With the aim of closely simulating all processes of real-world neural organoid experiments, we propose a neural organoid simulation framework using the elements mentioned in Section 3. It comprises of three main components, information encoding and Observation/Stimulation(O/S) array establishment, O/S information transmission mechanisms, organoid network architecture and learning strategy. Furthermore, we develop an organoid intelligence extension platform based on SNN to provide additional support for research in oraganoid-machine collaborative intelligence. The following subsections will go over the details of the framework.

### 4.1 information encoding and O/S array establishment

In typical neural organoid experiments using MEA, information is transmitted through electrical signals that interact with neurons via electrodes. Therefore, the primary task is to transform diverse natural signals into electrical signals with minimal loss. Simultaneously, the form of the O/S array(corresponding to the MEA) is determined depending on the requirements of different types of information. We present two encoding methods for two common input formats.

**Intensity Encoding.** Element values are utilized to represent information of fixed-size discrete matrix data (e.g. images), and are considered as intensities. Mapping the intensity to electrical currents enables the encoding of information. Excessive currents leads to the physiological demise of neurons in real-world experiments. So, it can identify a upper limit of external current, denoted as $I_{max}$. After obtaining the external current range $[0, I_{max})$, intensity values of matrix is normalized to fit within the range, providing the current stimulation intensity matrix. It is a common practice to set the size of O/S array as an integer multiple of the matrix size, allowing for a one-to-one or spaced correspondence between the positions of the matrix elements and the nodes in the O/S array. The schematic diagram of the intensity encoding is depicted in the first process line of the top-left subplot in Fig.3.

**Position Encoding.** In some studies investigating the intelligence in neural organoids (Kagan et al., 2022; Cai et al., 2023), the stimulation method is often straightforward. They use stimulation sites on the MEA to form their own specific patterns and represent information. This approach can represent simple pattern information, such as numbers, letters and geometric shapes. Although it appears overly simplistic compared to AI's information processing method, this method is widely utilized in neural organoids. Initially, the simple patterns are abstracted into discrete points on a 2D coordinate axis artificially, which is called "position encoding". An O/S array of appropriate size is generated based on the initial process, and the position encoding is mapped one-to-one into the corresponding positions inside the O/S array. Following the identification of the stimulation site, a preset stimulation pattern is applied, which is typically set at 20mA and 1Hz in real-world experiments. The schematic diagram of the position encoding is depicted in the second process line of the top-left subplot in Fig.3.

### 4.2 O/S information transmission mechanisms

During the O/S process, encoded input information is converted into temporal current/voltage signals. These signals are then applied to the O/S array and transmitted to the neural organoid network via each O/S node. While the neural organoid network learns the information and generates a response, neuron activity can also be read from the O/S nodes. From the perspective of common AI

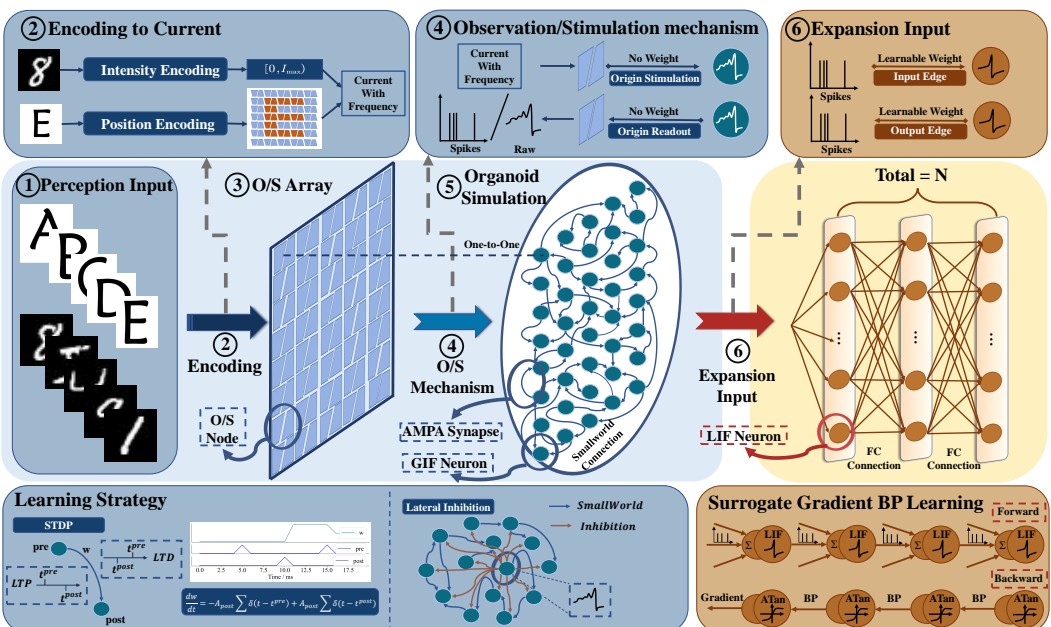

Figure 3: The overall architecture of the proposed method. Cold color background represents the NOSF, warm color background represents the intelligent expansion platform.

methods like perceptron, CNN and SNN, O/S array appears to serve as both the input and output layers. The weights exist between the input layer/output layer and hidden layers in traditional AI methods. These weights are vital for extracting initial features and ultimately classifying features. However, MEA and neurons are only connected for signal transmission in real-world experiments. There are no weights involved in the connections between O/S nodes and neurons inside the simulation framework's O/S information transmission processes, as shown in Subplot 4 in Fig.3. Compared to typical AI methods, there is a fundamental difference from the input and output layers in information transmission.

## 4.3 ORGANOID NETWORK ARCHITECTURE AND LEARNING STRATEGY

Before constructing the oganoid network, it is necessary to determine the correspondence between neurons and O/S nodes. We devised a simple and effective method where each O/S node corresponds precisely to a single neuron. This correspondence can efficiently monitor or stimulate each single neuron, which is also the ideal setup pursued in real-world experiments. The size of electrodes of the advanced MEA can be smaller than that of the neurons. Hence, a single electrode matching to a single neuron is functionally possible. Since the proliferation of neuron is typically not interfered during the organoid culture process, the growth direction is chaotic and the growth density is uneven. Our framework implements precise stimulation of neurons, resulting in a one-to-one correspondence that makes experiments simpler.

Once the O/S array's size is determined, the required number of GIF neurons is calculated based on the one-to-one correspondence. The small-world connection pattern is used to randomly generate the connection relationship between neurons, with AMPA synapses functioning for the connections. The oragnoid neural network architecture is displayed in the subgraph 5 in Fig.3. Under the basic configuration, only one single layer of neurons, referred to as the "2D model", adheres closely to the O/S nodes. An expansion method is also proposed for expanding more layers of neurons, referred to as

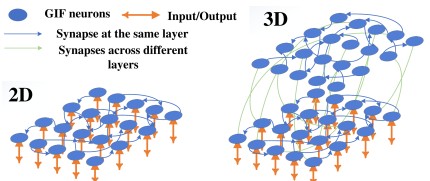

Figure 4: Differences between the 2D and 3D models

the "3D model". A 3D model with more layers is equivalent to duplicates of a single 2D model, with small-world connections used between layers. Fig. 4 illustrates the differences between the 2D and 3D models. Unlike traditional feedforward neural networks, the multi-layer organoid network does

not output responses from the outermost/last layer. Whether the model is 2D or 3D, the response is output from the nearmost/first layer of neurons adhered to the O/S node. In this form, organoid neurons learn and perceive information only in their interior which can be called as "oscillations".

After receiving information stimulation, the organoid neural network learns by enhancing the plasticity of the synapses to adjust the connection weight in the simulation framework. We utilize the modified Spike-Time-Dependent Plasticity (STDP) mechanism to adjust synaptic weights. STDP adjusts synaptic weights based on the timing of presynaptic and postsynaptic spikes, which is a typical unsupervised learning method consistent with Hebbian learning principles (Bi & Poo, 1998). Inspired by Paredes-Vallés et al. 2020 and Dong et al. (2022), the definition of STDP used in this work is:

$$\frac{\mathrm{d}w}{\mathrm{d}t} = \frac{1}{N_{batch}} \sum_{i=1}^{N_{batch}} \left( \eta_+ x_{pre}^{(i)} \delta_{post}^{(i)} - \eta_- x_{post}^{(i)} \delta_{pre}^{(i)} \right) \tag{3}$$

$$\frac{\mathrm{d}x_{pre}^{(i)}}{\mathrm{d}t} = -\frac{x_{pre}^{(i)}}{\tau_+} + \delta(t), \frac{\mathrm{d}x_{post}^{(i)}}{\mathrm{d}t} = -\frac{x_{post}^{(i)}}{\tau_-} + \delta(t) \tag{4}$$

where $N_{batch}$ is the number of samples in the corresponding batch, while $\eta_+$ and $\eta_-$ are constants representing the learning rates of the Long-Term Potentiation (LTP) and Long-Term Depression (LTD) mechanisms. $\delta_{pre}^{(i)}$ and $\delta_{post}^{(i)}$ indicate whether the pre-synaptic neurons of the $i$-th sample are releasing neurons at the current moment. $\tau_+$ and $\tau_-$ are time constants. To enhance competition among neurons and facilitate faster model convergence, we draw inspiration from the E-I balance network in Diehl & Cook (2015). Lateral inhibition and weight attenuation mechanisms are introduced to further optimize this model. Due to lateral inhibition, when a neuron fires a spike, it suppresses neighboring neurons from firing. This is equivalent to the presence of inhibitory synapses between the firing neuron and its neighbors by reducing their membrane potential. Besides, weight attenuation mechanism is introduced to restrict the weight range and prevent some neurons from exhibiting harmful levels of competitiveness. When synapses are inactive for an extended period, their weights attentuate, which means positive weights will decrease and negative weights will increase.

### 4.4 INTELLIGENT EXPANSION PLATFORM

We develop an intelligent expansion platform based on SNN, cooperating with the simulation framework. The goal of intelligent expansion is to utilize advanced AI technology to compensate for organoid intelligence's weakness and further look into the potential of organoid-machine intelligence. In SNN, information is transmitted via spike mechanism that aligns with the output of the organoid simulation framework. Therefore, SNN is employed to establish the intelligent expansion platform, which includes Leaky Integrate and Fire (LIF) neurons that are described as:

$$\frac{\mathrm{d}V(t)}{\mathrm{d}t} = \frac{1}{\tau}(RI - (V(t) - V_{rest})) \tag{5}$$

where $V$ is membrane potential, $V_{rest}$ is resting potential, $R$ is membrane resistance, and $I$ is external current. Once $V$ exceeds the threshold $V_{th}$, a spike will be fired. LIF neurons are the most widely used neuron model in the SNN domain becuase of their tremendous computational simplicity. The platform uses basic linear layers as main architecture, and the number of layers can be adjusted according to experimental requirement. The output of the two-dimensional O/S array is flattened into a one-dimensional vector, which serves as the input for the linear SNN. As shown in the bottom right subfigure of Fig.3, the platform uses the supervised surrogate gradient backpropagation algorithm for network learning. Supervised learning can improve the ability of certain tasks such as object recognition, in a specific direction.

## 5 EXPERIMENTS

### 5.1 SIMULATION SIMILARITY EVALUATION

**Benchmark.** The first benchmark is proposed for our organoid simulation framework in this section, which comprises a set of real-world organoid experiment data and evaluation metrics. In the real-world organoid experiment, a cluster of 200-day neural organoid adheres to the 8x8 MEA. The

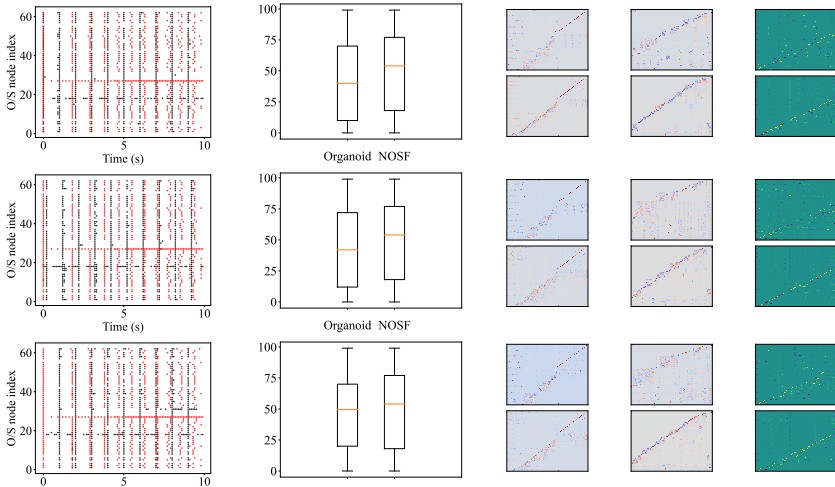

Figure 5: Comparison between real-world experiment organoid data and simulation framework results. In order from left to right, the figures show: the schematic of firing spike, the boxplot of spike firing timestamps, the results of SVD, and the results of QR. In order from top to bottom, the figures illustrate three sets of the real-world organoid data and the simulation framework results.

MEA electrodes are used to draw shapes with semantic meaning, such as numbers or geometric figures. Regular stimulation is applied at 500mV and 1Hz. Each round of stimulation lasts for 10 seconds, and three rounds are conducted to obtain three sets of data. We present evaluation metrics from different perspectives to assess the resemblance between real-world organoid experiment data and results of the simulation framework. From the perspective of mathematical analysis, we introduce two types of metrics, including matrix analysis and statistics analysis. The spikes recorded by MEA within 10s is regard as a matrix for matrix analysis. We employ some metrics about matrices to measure this similarity, such as singular value decomposition (SVD), QR decomposition and spectral norm. SVD and QR decomposition are conventional techniques for extracting matrix features in matrix analysis. Spectral norm is used to measure the matrix stability, and we consider that having similar stability is required for measuring similarity. For statistics analysis, the average firing rate is used to measure the total number of spikes. Subsequently, we determine the timestamps of spike firing and calculate the mean, variance and average maximum/minimum firing intervals. From the perspective of bionics, we introduce two metrics: inherent characteristics of neurons and Hebb Learning Rule (Hebb, 1949). The inherent characteristics of neurons include extensive resting state and correlation spikes. Hebb Learning Rule suggests that when one neuron repeatedly stimulates another neuron, the connection strength between them will enhance. Check the Appendix A.2 for further information on evaluation metrics.

*BrainPy* (Wang et al., 2022) is utilized to construct our framework. We set the number of GIF neurons to 64 and the $K$ neighbors of small-world connection to 5. Outstanding results are obtained in the displayed metrics as shown in Fig.5. The spike firing diagram shows that the real-world organoid data (black dots) and the simulation framework results (red dots) exhibit substantial parallelism, with both having a fixed burst period (vertical line) and a highly competitive neuron (horizontal line). We make a boxplot of all the spike firing time to visualize the spike distribution throughout the whole time period. The similarity can be clearly defined by the size of the box and the position of the quadrant line. The slope and width of diagonal lines, which shows a considerable resemblance, must be observed for the results of the two types of decomposition. Table 1 illustrates the high similarity between the real-world experiment organoid and the simulation framework results in mathematical analysis metrics. The small difference in the specific values of the first two rows reflects the aforementioned similarity. Each value in the last four lines represents a specific time value in seconds. Strong similarity is proved by calculating the absolute error between these values. For instance, the difference between 2.33 seconds and 2.49 seconds is only 0.16 seconds, which is relatively minor considering the entire time period (10 seconds). Check Appendix A.2 for more details . Regarding the bionic metric, the red region in the first column of Fig.5 illustrates that only a small portion of the entire time generates spikes, indicating that the system is extensively resting. Simultaneously, the GIF model reflects non-spontaneous correlation spikes that generate in response to external input.

Table 1: Comparison of the real-world organoid data and the simulation framework results with mathematical analysis metrics.

| Metric | Group 1 | | Group 2 | | Group 3 | |
|---|---|---|---|---|---|---|
| | Organoid | NOSF | Organoid | NOSF | Organoid | NOSF |
| Spectral norm | 23.32 | 21.00 | 22.87 | 20.62 | 21.72 | 19.95 |
| Average firing rate | 6.16 | 6.41 | 6.31 | 6.35 | 5.84 | 6.39 |
| Average firing time | 2.33 | 2.49 | 2.44 | 2.48 | 2.56 | 2.49 |
| Firing time variance | 1.76 | 1.56 | 1.77 | 1.56 | 1.66 | 1.56 |
| Average maximum firing interval | 0.74 | 0.90 | 0.85 | 0.90 | 0.86 | 0.90 |
| Average minimum firing interval | 0.40 | 0.35 | 0.20 | 0.36 | 0.46 | 0.58 |

Table 2: Accuracy of classifying 0 and 1 under different hyperparameters.

| Model | GIF | AMAP | | | Small-world | Training | | Result(%) |
|---|---|---|---|---|---|---|---|---|
| | $\tau$ | $\alpha$ | $\beta$ | $T\_dur$ | $K$ | $delay$ | $sim\_T$ | |
| 2D | **4** | 0.96 | 0.6 | 0.01 | 5 | 12 | 60 | 94.70 |
| | 5 | **0.8** | 0.6 | 0.01 | 5 | 12 | 60 | 94.23 |
| | 5 | 0.96 | **0.8** | 0.01 | 5 | 12 | 60 | 92.34 |
| | 5 | 0.96 | 0.6 | **0.05** | 5 | 12 | 60 | 95.56 |
| | 5 | 0.96 | 0.6 | 0.05 | **10** | 12 | 60 | 64.26 |
| | 5 | 0.96 | 0.6 | 0.01 | 5 | **16** | 60 | 94.99 |
| | 5 | 0.96 | 0.6 | 0.01 | 5 | 12 | **100** | 94.99 |
| | **5** | **0.96** | **0.6** | **0.01** | **5** | **12** | **60** | **96.80** |
| 3D | **6** | 0.96 | 0.6 | 0.01 | 5 | 12 | 60 | 92.39 |
| | 5 | 0.96 | 0.6 | 0.01 | **10** | 12 | 60 | 54.00 |
| | 5 | 0.96 | 0.6 | 0.01 | 5 | **8** | 60 | 90.07 |
| | **5** | **0.96** | **0.6** | **0.01** | **5** | **12** | **60** | **94.52** |

Since the Hebbian Learning Rule is one of the characteristics of STDP, our framework conforms to this rule.

## 5.2 SIMULATION INTELLIGENT EVALUATION

Some previous works(Zheng et al., 2022) have demonstrated their own intelligence by letting neural organoids classify simple patterns. Similarly, we classify digits 0 and 1 in MNIST to ascertain whether the simulation framework exhibits consistency in own intelligence performance as real-world neural organoids. Since various hyperparameters may influence the result, we include adjustments to them into the experimental process to investigate some critical part of our framework.

We employ a voting mechanism to determine the classification accuracy. Firstly, we assigned each neuron a category based on its highest average firing rate for the digits 0 and 1 across training set samples. Subsequently, the predicted digit for each image is determined by averaging the spike count of each neuron per class and selecting the class with the highest average spike count. As shown in Table 2, the results fully prove the own intelligence of this framework and the correlation between the own intelligence and the real-world organoid experiments. For both the 2D and 3D models, we highlight the best results in bold red text. Meanwhile, we discover that the appropriateness of particular hyperparameter settings will greatly affect the performance of the framework. Therefore, we finetuned each hyperparameter (black with bold and underlined in Table 2) based on the optimal result and evaluated the subsequent changes in the accuracy rate. Since 2D models have sparser connections compared to 3D models, most hyperparameter adjustments will not result in a significant decrease in accuracy (such as $\tau$ and $delay$). As we increase the value of $K$ (from 5 to 10), the network becomes denser. The initial weights not being adjusted will result in the generation of a large number of spikes, making it difficult for STDP to learn additional features. As a result, the correct rate will decline sharply. Since 3D models are denser, little changes in hyperparameters might result in significant accuracy decreases. For example, decreasing the value of $delay$ will greatly reduce the efficiency of STDP learning for 3D models with a larger number of synapses, resulting in a significant decrease in accuracy. Check the Appendix A.3 for more details related to the experimental setting.

## 5.3 Intelligent expansion evaluation

Consistent with the methods of classifying 1 and 0, the organoid simulation only achieves 31.45% accuracy in classifying complete Mnist, which is consistent with the methods of classifying 0 and 1. The output of the simulation framework is used as the input data for SNN and the accuracy improvement with minimal computational burden is tested. Specifically, we set up an single linear layer SNN with 28×28 neurons. Subsequently, We flatten the 2D data observed by the simulation framework into 1D data and feed it into the SNN linear layer, resulting in an enhancement of the classification accuracy to 91.64%.

It verifies the feasibility of the intelligent expansion platform, achieving the goal of intelligent supplementation and collaboration with organoids. In addition, we also conduct a comparison experiment by introducing ANN and extra SNN with the same structure but a different input scheme. We also investigated whether extra layers would improve performance, experimenting with 1 layer, 2 layers, and 3 layers config-

Table 3: The accuracy comparison of different algorithms and different layers on Mnist.

| Method | 1 layer | 2 layers | 3 layers |
|---|---|---|---|
| ANN | 92.79% | 97.89% | 98.34% |
| SNN (Poisson Encoding Mnist) | 92.71% | 98.34% | 98.69% |
| SNN (Simulation Output) | 91.64% | 97.60% | 97.96% |

urations. The results, as shown in Table.3, demonstrate that the accuracy of the organoid simulation+SNN intelligent expansion solution is comparable to the pure AI method of ANN and SNN, with an accuracy difference of less than 1%, proving the superiority of the intelligent expansion platform.

## 5.4 Discussion and Limitation

When the simulation framework's experiment results are compared to the mainstream recognition methods in the field of AI, the simulation framework appears to only produce relatively mediocre results. Nevertheless, it is worth noting that our aspiration is to reproduce the performance of real-world neural organoids using the simulation framework, rather than pursuing the optimal performance. The intelligence exploration of current real-world neural organoid is relatively primitive, far from reaching human intelligence. The simulation framework also performs some basic tasks and obtains feedback that aligns with real-world organoids. There are still some limitations in our work. In most real-world organoid experiments, a one-to-one correspondence between neurons and MEA electrodes cannot be achieved. An electrode frequently captures responses from multiple neurons, making it difficult to determine the electrical signal state of a specific single neuron. On the other hand, the neuron and synapse model used in our framework is still not detailed enough, and a gap from the dynamic processes of real-world neurons still exists. In terms of learning, real-world neurons tissues enhance intelligence by improving synaptic plasticity and growing new neurons. However, due to the unclear biological growth mechanism, our framework only implements the synaptic plasticity and cannot support real-time neuron expansion. We are focusing on proposing a novel path, and trying to further explore intelligent expansion with other AI researchers.

## 6 Conclusion and future work

In this paper, we present the first neural organoid simulation framework with a intelligence expansion platform to save costs of repeated real-world experiments, resulting in a outstanding performance. In the future, complex correspondence between neurons and O/S nodes, detailed biological modeling, and neuron proliferation simulation are attractive directions to pursue.

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

# A APPENDIX

## A.1 PRELIMINARIES

### A.1.1 GENERALIZED INTEGRATE-AND-FIRE NEURON

In the realm of bionic computational modeling for neurons, numerous models have been proposed, including the Hodgkin-Huxley (H-H), Morris-Lecar , Izhikevich, Leaky Integrate-and-Fire (LIF) and others. Different models have different emphases, for example, H-H model focuses on ion channel modeling, and LIF model focuses on release mode reduction and computational simplicity. In our framework, the focus for neurons is on precise spike delivery and accommodating various delivery modes (e.g. burst, bistability, etc), rather than specific ion channels. For this reason, we opt for the Generalized Integrate-and-Fire model as the neuron component within the framework. This model offers a detailed representation of spike delivery while striking a balance between performance and computational complexity. Specifically, we use the scheme of (Mihalaş & Niebur, 2009) and (Teeter et al., 2018), which can be described as:

$$\frac{\mathrm{d}I_j(t)}{\mathrm{d}t} = -k_j I_j(t); \quad j \in [1, N] \tag{6}$$

$$\frac{\mathrm{d}V(t)}{\mathrm{d}t} = \frac{1}{\tau}(RI + R\sum_j I_j(t) - (V(t) - V_{rest})) \tag{7}$$

$$\frac{\mathrm{d}V_{th}(t)}{\mathrm{d}t} = a(V(t) - V_{rest}) - b(V_{th}(t) - V_{th\infty}) \tag{8}$$

where $V$ is membrane potential, $V_{th}$ is instantaneous threshold, $V_{rest}$ is resting potential, $R$ is Membrane resistance, $I_j$ are an arbitrary number of internal currents and $I$ is external current. $k_j, \tau, a, b$ are control arguments respectively. Once $V$ exceeds the threshold $V_{th}$, a spike will be fired. Meanwhile, $V$ will be reset to the resting potential $V_{reset}$, $V_{th}$ will follow $V_{th} = \max(V_{th_{reset}}, V_{th})$ where $V_{th_{reset}}$ is reset threshold, and $I_j(t)$ will be defined as $I_j(t) = R_j \times I_j(t) + A_j$ where $A_j$ is constant parameter.

### A.1.2 AMPA SYNAPSES

In bionic computational modeling of synapses, they are primarily categorized into two main types based on information transmission: chemical synapses and electrical synapses. Electrical synapses exhibit high speed transmission but lack regulation and have limited information transmission capacity. Conversely, chemical synapses are more flexible and their plasticity contributes to explaining the mechanisms of learning and memory within the nervous system to some extent. Hence, establishing a simulation framework using chemical synapses is essential. Two common chemical synaptic models are based on the AMPA receptor and the NMDA receptor. Since AMPA receptors have a fast response, making them more suitable for our framework implementation, we opt for the AMPA receptor model as the foundational synaptic model for the framework. Specifically, we adopt the approach employed by (Vijayan & Kopell, 2012):

$$\frac{\mathrm{d}g}{\mathrm{d}t} = \alpha[Glu](1 - g) - \beta g \tag{9}$$

$$I = Gg(V - E) \tag{10}$$

where $g$ represents the concentration of AMPA receptors (which, to some extent, also represents the probability of channel opening), $\alpha$ signifies the binding rate constant between AMPA receptors and glutamic acid, while $\beta$ represents the dissociation rate constant. $[Glu]$ represents the concentration of the external neurotransmitter. Under normal circumstances, it is considered to be a hopping process. When the presynaptic neuron firing, it immediately changes to $T_0$, and continues for $T_{dur}$ and then quickly changes to 0. $I$ represents the postsynaptic current (PSC), $G$ stands for the maximum conductance, $V$ denotes the postsynaptic neuron membrane potential, and $E$ represents the reversal potential.

### A.1.3 Small-world network

Small-world network (Watts & Strogatz, 1998) is a distinctive type of complex network that is prevalent in reality and holds substantial practical significance. Specifically, after extensive research, it is observed that many large-scale neural networks within the brain, including those in the visual system and the brain stem, exhibit the characteristics of a small-world topology. Furthermore, at the level of computational modeling, small-world networks exhibit favorable short-term memory characteristics. Hence, in comparison to other topological structures like complete graph networks and probabilistic connection networks, the small-world network possesses greater biological relevance and practical significance, making it a superior choice for the network structure of this framework. In terms of the small-world network, it is characterized by:

1) Build a regular ring lattice with $N$ nodes, each connected to $K$ neighbors.

2) Reconnect each edge of each node in sequential order with a probability of $p$ and randomly select the target nodes for reconnection.

### A.2 EVALUATION METRICS

To assess the similarity between real-world organoid data and simulation framework results, we propose evaluation metrics from different perspective.

From the perspective of mathematical analysis:

- For matrix analysis, we regard the spikes recorded by MEA within 10s as a 100 x 64 matrix. Here, the 100 dimensions represent time, and the 64 dimensions correspond to 64 O/S nodes. Consequently, the primary focus lies in analyzing the crucial characteristics of these matrices. If these crucial characteristics are similar, it demonstrates a strong similarity between real-world organoid data and simulation framework results. In matrix analysis, there are two classical approaches for extracting features: Singular Value Decomposition (SVD) and QR decomposition. For SVD, it can be expressed as:

$$A = U\Sigma V^T$$

  where U and V matrices are SVD results, which can be met by computing the eigenvalues and eigenvectors of $AA_T$. For QR decomposition, it can be expressed as:

$$A_{m \times n} = Q_{m \times n} R_{n \times n}$$

  where $Q$ is a standard orthogonal vector group matrix and $R$ is a positive upper triangular matrix. We can use Schmidt orthogonalization to get $Q$ and use $R = Q^T A$ to get R. Simultaneously, we extract and compare the spectral norm which provides a numerical representation of the similarity between real-world organoid data and simulation framework results. Spectral norm is the largest singular value in SVD.

- From a statistical perspective, certain key statistics provide insights into the overall characteristics of real-world organoid experiment data and results of the simulation framework, including measures such as mean and variance. When they are substantially similar, these statistics can also imply some degree of similarity. In this context, we examine the firing rates of all O/S nodes over the entire 10-second period. We count the spikes and divide it by the total time. We also calculate the mean and variance for the timestamps of spikes firing of each O/S node, as well as determine the maximum and minimum firing intervals, which reflect the activity levels of neurons. We record the timestamp of each spike and calculate the mean and variance, which helps in depicting the overall spike distribution throughout the entire time period. Simultaneously, for each O/S node's timestamps, we compute both the maximum and minimum firing intervals and then calculate the average. This provides insights into the total neuronal excitability. These statistical analyses allow us to evaluate the similarity from a statistical standpoint.

From a bionics perspective, we introduce three metrics to assess the bionic characteristics of this framework. These metrics include: (i) Extensive Resting State. It refers to the observation that neuron clusters spend most of their time in a resting state. This behavior is the natural resting state observed in biological systems. (ii) Correlation spikes. This metric focuses on the generation of

spikes as a result of interactions with other spikes, rather than spontaneous generation. (ii) Hebb Learning Rule (Hebb, 1949). The Hebb learning rule suggests that when one neuron repeatedly stimulates another neuron, the connection strength between them enhance. This rule is widely found in existing biological systems.

## A.3 HYPERPARAMTERS

Regarding certain crucial hyperparameters in the experiment, we provide explanations for each of them below:

- In the GIF model, $\tau$ governs the scale of change in neuron membrane potential. A large value for $\tau$ signifies a small growth rate of the membrane potential, which is evident in the dynamic equation of the GIF neuron model.

- $\alpha$ controls the rate of change in synaptic conductance. In biological terms, it represents the binding rate of the AMPA receptor to glutamic acid. In this framework, a larger $\alpha$ value implies that AMPA synapses will require more time to stabilize.

- $\beta$ also controls the rate of change in synaptic conductance. In contrast, in biological terms, it represents the unbinding rate of the AMPA receptor to glutamic acid. In this framework, a larger $\beta$ value implies that AMPA synapses can stabilize more quickly. Besides, it's imperative that $\alpha$ is greater than $\beta$.

- $T\_dur$ plays a crucial role in governing the growth scale of AMPA neurons from a state of rest to stability. A larger $T\_dur$ leads to more stable AMPA usage.

- $K$ represents the number of synapses connected from one neuron to adjacent neurons when constructing the small-world connection. In fact, it reflects the sparseness of the network. A larger $K$ will result in a denser network.

- $delay$ signifies the magnitude of network weight attenuation. An appropriate $delay$ can help the model to fit faster. If $delay$ is set to be too large, the synaptic weight in the model will be excessively low. Furthermore, if there is reduced neural activity in the network, meaning fewer spikes are released, the network will deteriorate rapidly. In such a scenario, the weight of each synapse will decrease swiftly, ultimately leading to the "deactivation" of all neurons. When the parameter $delay$ is set to be too small, it may not effectively suppress the weight increase caused by STDP. Conversely, if the weight increasing is too large, it can result in high-intensity network activity. In extreme cases, every neuron may continuously release spikes, which is clearly contrary to the real-world organoid mechanisms.

- $sim\_T$ represents the duration of a simulation. In the training set, this data essentially corresponds to the time required for STDP to learn. Selecting an appropriate value for $sim\_T$ is crucial. If it is set too small, STDP may not learn enough features, resulting in poor results. On the other hand, if it is set too large, $delay$ may not be sufficient to restrain the increase in STDP weight, which results in excessive synaptic weight growth within the network.

In simulation intelligent evaluation, we use the 2D and 3D models described above (Section 4). We employed 57x57 neurons to create a two-dimensional plane, while also evenly selecting 28x28 neurons to serve as O/S nodes. For 3D model, it is essentially a 2D model with two layers. Furthermore, the layers are interconnected by small-world connection. In terms of intensity encoding, we first normalize the gray value of the image and then encode it using $I_{max} = 64$.

