# OpenReview forum: "To Simulate Neural Organoid: A Framework and A Benchmark based on AI"
_ICLR.cc/2024/Conference — Submitted to ICLR 2024_

### Official Review · Reviewer_qTxr · 2023-10-29

**Soundness:** 3 good
**Presentation:** 2 fair
**Contribution:** 1 poor
**Rating:** 3
**Confidence:** 5

**Summary:**

The authors use the Brainpy simulation package to run spiking network simulations of brain organoids, and simulations of organoids connected to artificial networks.  They train a classifier on simulated organoid output, which can distinguish two digits from MNIST.

**Strengths:**

The simulation results match measures of organoid activity by some criteria.

**Weaknesses:**

It is not clear what the motivation fort this work is. Organoids are not a major model of neural computation within experimental neuroscience: very few experimental neuroscientists believe they have anything like the computational power of actual brains.  (The authors use the word “intelligence” but few would claim organoids possess intelligence as the word is usually understood.) The authors own results suggest that random networks perform similarly to organoids.  If so, then what do we learn from organoids that we do not already know about random networks?

**Questions:**

The paper would have been stronger if it addressed specific hypotheses. Some examples would be: how does the behavior of organoids differ from random networks?  Does the STDP rule make the networks more or less than actual organoids?  How does organoid activity differ to in vivo brain activity, and what simulation parameters best model each one?

---

### Official Review · Reviewer_WMhb · 2023-10-31

**Soundness:** 3 good
**Presentation:** 2 fair
**Contribution:** 1 poor
**Rating:** 3
**Confidence:** 4

**Summary:**

This manuscript provides a simulation framework for cultured neuron experiments placed on top of a multi electrode and recording array. They model cultured neurons as a set of generalized integrate and fire cells with AMPA-based learning and small world structure. They apply patterned mutlielectrode stimulation to either all the  model neurons with a 1:1 mapping, or a fraction of the model neurons if they assume a 3D culture. Feed output of the network as a spiking neural network to a classifier. They compare the results of the framework with data from a real organoid system that is placed on a multi-unit stimulation and recording array. They find comparable firing rates when both systems are strongly driven by patterned stimulation, as well as several other metrics such as timing variance. They then place a spiking neural network at the output of the simulated neural network and train it on a binarized detection task or mnist, and find comparable accuracy to pure ANN approaches.

**Strengths:**

* Figures explaining the experiments were clear, and I think the method is well described.

* Simulation methods for modeling organoids are important.

**Weaknesses:**

* The language describing results and nomenclature is a bit sensational “resulting in remarkable simulation results”, and overall there is a lot of nonstandard terminology used in the paper described more below.

* Even if this exact implementation of a neural network simulation environment is novel, its contribution is more in assembling off-the-shelf components and may be more suitable for a publication venue where the experimental results are more emphasized. The results are more around simulations of real neural networks which isn’t a main focus of ICLR.

* The comparisons with real neural networks in Figure 5 were difficult to understand – it is unclear what the axes and units are for several of the plots (eg SVD and QR) and it is unclear what the real and simulation values are for the spiking schematics. The boxplots appear significantly different. In addition, I am not so impressed by demonstrations of firing rate agreements when the entire network is being strongly and synchronously driven.

**Questions:**

* Unclear what intelligence expansion is .

* “Neural organoids culture and intelligence.” Section conflates organoids with retinal explants, acute cultures of dissected brain regions, cultured neurons, cultured iPSC cells and full 3D organoids. This is important because by claiming novelty the paper has to distinguish itself from modeling of these other domains where there has been tremendous work modeling the interaction between cells on a dish and patterned multi-unit excitation.

* “SNN algorithms are renowned for their robustness, biological interpretability, and energy efficiency in the Artificial Intelligence community.” This seems incorrect.

* “Specifically, after extensive research,it has been proved that many large-scale neural networks within the brain, including those in the visual system and the brain stem, exhibit the characteristics of a small-world topology.” Needs citation.

* Experiments of the organoids are only cursorily described

---

### Official Review · Reviewer_kTjU · 2023-11-01

**Soundness:** 2 fair
**Presentation:** 1 poor
**Contribution:** 2 fair
**Rating:** 3
**Confidence:** 2

**Summary:**

The authors propose a framework for simulating neural organoids.  This framework is based on the Mihalas & Niebur GLIF model (that the authors describe using "GIF"), an AMPA receptor model, and Watts & Strogatz small world networks.  They include an "intelligent expansion platform" to "compensate for the organoid intelligence's weakness and further look into the potential of organoid-machine intelligence."  This expansion appears to be a network of LIF neurons responding to input from the organoid model.  They test their model on real-world organoid experimental data.

**Strengths:**

A modeling framework for neural organoids would be a useful tool.  Code is supplied to supplement the limited (to this reader) descriptions of the model construction.

**Weaknesses:**

The paper reads somewhat opaquely (this could be partially due to my being less familiar with some of the subject matter of the paper, but I think there is much room for improvement in the clarity of the writing in any case.  There are many awkward idioms that make it difficult to interpret at times.).

It is not clear what the origin of the real world data is.  Did the authors collect it themselves?  Is there a reference?  Where did it come from?

Similarly, the descriptions of results comparing the model with real-world data seem lacking, with a few rows of unlabeled plots and an incomplete caption describing the plot.  What is the far right of Figure 5?  I assume top and bottom on the right of each row are Organoid and NOSF, but which is which?  Are we comparing them simply visually, or in a quantitative way?

Perhaps it is my ignorance of the experimental side here, but the metrics is Table 1 seem fairly each to match with neural models (such as specifically firing rate and variance).  What parts of your model are necessary for your results?

I'm also unsure of what to make of the classification tests.  What's the goal here?  Surely it isn't raw performance, as that would be a low bar for this test.

I can see how the model is constructed and what it does (more or less), but I'm left wondering what metrics would be used to determine whether it does its job well.

**Questions:**

Is it possible to be more explicit about what a "good" organoid model would do?  What metrics would tell you you had been successful?

---

### Official Review · Reviewer_V5kH · 2023-11-01

**Soundness:** 2 fair
**Presentation:** 1 poor
**Contribution:** 1 poor
**Rating:** 3
**Confidence:** 4

**Summary:**

This work sets out to model—and supplement—neural organoids with in silico models, namely, spiking neural networks (SNN). The organoid network model presented is an SNN with GLIF neurons, AMPA synapses, and small world connectivity with lateral inhibition, and synaptic weights can be tuned via STDP rules. The authors show that network simulations have similarities to recorded organoid spiking activity. Furthermore, the spiking network model can perform binary classification with good accuracy, and can be further supplemented by a spiking network trained via surrogate backprop to perform the full 10-class MNIST task.

**Strengths:**

The work is an interesting combination of in vitro and in silico network models, with the aim of integrating biological and artificial intelligence based on neuronal networks. Additionally, various hyperparameters values are explored, adding confidence to the robustness of the classification task results. Also, the authors explicitly acknowledged a range of limitations to their study, which is always nice to see.

**Weaknesses:**

At its core, this work presents a spiking neural network model that receives spatially organized input and is able to perform MNIST classification, with and without the help of additional SNN layers that are trained via surrogate gradient descent. All of the above have been demonstrated previously, much of it is also cited by the authors, so they do not represent significant contributions.

Furthermore, these technical contributions have little to do with neural organoids: Figure 5 shows that simulated network outputs have similarities to organoid output, thus representing evidence that the in silico model is a model for the organoid in vitro. However, this evidence is unconvincing. Spiking network models, under many parameter configurations, tend to exhibit synchronized network bursts, which the organoids also happen to exhibit naturally. Thus, the fact that they both burst at similar (and a very broad) firing rate is not convincing evidence that the model matches the organoid. More importantly, there are many network model parameters that alter its behavior. The authors explore some of these in the context of classification performance, but not in how well the model matches the organoid—indeed, mechanistic parameter identification based on real data is non-trivial, and significantly more evidence is required to show that the model is similarity to the in vitro system.

Finally, I personally found the text to be quite difficult to understand, not due to low-level grammatical issues, but that it is often written in a manner that does not convey much scientific detail. For example, the abstract reads more like a press release, with very little details regarding the methods and key results, but is rather more focused on conveying the importance of the problem. While I agree that this avenue of research is interesting, it’s ultimately framed inappropriately (in my opinion), and in the present state, the work falls very short of the stated goal of AI-organoid symbiosis.

**Questions:**

- are the similarity measures between organoids and simulated networks sensitive to network parameter values? Many of the parameters can take on a range of values in real biological systems, how do the authors choose these values?
- what are the variances for the values in Table 1?
- How do the organoids behave when given the same MNIST inputs? Do they respond similarly to the simulations?
- Figure 3 is very full, and most of the details are illegible while the caption is not very informative
- Figure 5, what are the last 3 columns? And they don’t look very similar between organoids and simulations.
- along with the “hype-inducing” abstract, the paper uses many words for which I’m not sure I know the meaning for, e.g.: advanced neural computing models, significant consumption, intelligent expansion, organoid intelligence expansion platform, collaborative intelligence

---

### Meta-Review · Area_Chair_BjPw · 2023-12-03

**Metareview:**

The paper proposes to use organoids to improve AI-- the reviewer had multiple concerns about the paper, both including its contributions and its clarity, and all reviewers voted for not accepting the paper. We do hope that their reviews will help the authors in improving their study.

**Justification For Why Not Higher Score:**

This was a unanimous, clear decision.

**Justification For Why Not Lower Score:**

Not sure what 'lower score' than reject would be.

---

### Decision · Program_Chairs · 2024-01-16

Reject